# Function of miR825 and miR825* as Negative Regulators in *Bacillus cereus* AR156-elicited Systemic Resistance to *Botrytis cinerea* in *Arabidopsis thaliana*

**DOI:** 10.3390/ijms20205032

**Published:** 2019-10-11

**Authors:** Pingping Nie, Chen Chen, Qian Yin, Chunhao Jiang, Jianhua Guo, Hongwei Zhao, Dongdong Niu

**Affiliations:** 1College of Life Sciences, Zaozhuang University, Zaozhuang 277160, China; ppnie@uzz.edu.cn; 2College of Plant Protection, Nanjing Agricultural University, Nanjing 210095, China; 2016102047@njau.edu.cn (C.C.); chjiang@njau.edu.cn (C.J.); jhguo@njau.edu.cn (J.G.); hzhao@njau.edu.cn (H.Z.); 3Institute of Botany, Jiangsu Province and Chinese Academy of Sciences, Nanjing 210014, China; yinqian@cnbg.net

**Keywords:** miR825 and miR825*, *Bacillus cereus* AR156, induced systemic resistance, *Botrytis cinerea* B1301, plant innate immunity

## Abstract

Small RNAs function to regulate plant defense responses to pathogens. We previously showed that miR825 and miR825* downregulate *Bacillus cereus* AR156 (AR156)-triggered systemic resistance to *Pseudomonassyringae* pv. *tomato* DC3000 in *Arabidopsis thaliana* (*Arabidopsis*). Here, Northern blotting revealed that miR825 and miR825* were more strongly downregulated in wild type *Arabidopsis* Col-0 (Col-0) plants pretreated with AR156 than in nontreated plants upon *Botrytis cinerea* (*B. cinerea*) B1301 infection. Furthermore, compared with Col-0, transgenic plants with attenuated miR825 and miR825* expression were more resistant to *B. cinerea* B1301, yet miR825- and miR825*-overexpressing (OE) plants were more susceptible to the pathogen. With AR156 pretreatment, the transcription of four defense-related genes (*PR1*, *PR2*, *PR5*, and *PDF1.2*) and cellular defense responses (hydrogen peroxide production and callose deposition) were faster and stronger in miR825 and miR825* knockdown lines but weaker in their OE plants than in Col-0 plants upon pathogen attack. Also, AR156 pretreatment caused stronger phosphorylation of MPK3 and MPK6 and expression of *FRK1* and *WRKY53* genes upon *B. cinerea* B1301 inoculation in miR825 and miR825* knockdown plants than in Col-0 plants. Additionally, the assay of agrobacterium-mediated transient co-expression in *Nicotiana benthamiana* confirmed that *AT5G40910*, *AT5G38850*, *AT3G04220*, and *AT5G44940* are target genes of miR825 or miR825*. Compared with Col-0, the target mutant lines showed higher susceptibility to *B. cinerea* B1301, while still expressing AR156-triggered induced systemic resistance (ISR). The two-way analysis of variance (ANOVA) revealed a significant (*P* < 0.01) interactive effect of treatment and genotype on the defense responses. Hence, miR825 and miR825*act as negative regulators of AR156-mediated systemic resistance to *B. cinerea* B1301 in *Arabidopsis.*

## 1. Introduction

Plants are equipped with sophisticated immune response mechanisms to resist pathogen attack [1,2]. Pattern recognition receptors (PRRs) constitute the first line of plant defense against pathogens by recognizing conserved pathogen-associated molecular patterns (PAMPs), resulting in PAMP-triggered immunity (PTI). The components of PTI include mitogen-activated protein kinase (MAPK) activation [3], defense-related gene expression, and callose deposition [2,4,5]. On the other hand, many pathogens secret multiple specific effectors to inhibit PTI in host plants [6,7,8], which have developed the second line of defense comprising resistance (R) proteins that target corresponding pathogen effectors, resulting in effector-triggered immunity (ETI) [9]. ETI causes hypersensitive response (HR) at the infected site to inhibit the growth of biotrophic pathogens [1,2].

Induced disease resistance in plants is effective in controlling infections of a wide variety of pathogens (bacteria, fungi, and viruses), as well as insect herbivores [10,11,12]. Systemic acquired resistance (SAR) and induced systemic resistance (ISR) are two types of well-studied induced resistance [13], both resulting in defense responses in local and systemic tissues [10,14,15]. Usually, plants express SAR when infected with a necrotizing pathogen [14], while ISR is triggered by some beneficial rhizobacteria, including plant-growth-promoting rhizobacteria (PGPR), such as *Pseudomonas* and *Bacillus* [10,16,17]. A large number of plant species have been found to express ISR, including tomato, rice, tobacco, cucumber, bean, and the model plant *Arabidopsis thaliana*. By expressing ISR, plants can effectively resist the attack of pathogenic microorganisms (bacteria, fungi, and viruses), as well as insect herbivores [12,17,18]. Rhizobacteria elicit ISR through priming; upon pathogen attack, plants in the primed state mount fast and strong cellular defense responses comprising cell-wall reinforcement [19], oxidative burst [20], expression of defense-related enzymes [21], and accumulation of secondary metabolites [22].

The phytohormones salicylic acid (SA), jasmonic acid (JA), and ethylene (ET) play essential roles in expressing SAR or ISR [23]. In SAR-expressing plants, levels of endogenously synthesized SA increase systemically, resulting in enhanced expression of the genes that encode pathogenesis-related (PR) proteins, such as PR1, PR2, and PR5 [10,24,25]. In contrast, ISR is activated through the JA/ET-dependent signaling pathway, involving plant defensin 1.2 (PDF1.2) [18,26]. For instance, the ISR induced by *Pseudomonas fluorescens* WCS417r in *Arabidopsis* is dependent on the JA/ET signaling pathway and NPR1 [27], as is the ISR elicited by *Bacillus cereus* AR156 (AR156) against *Botrytis cinerea* (*B. cinerea*) B1301 in *Arabidopsis* [28]. However, some rhizobacteria, including PGPR, were shown to trigger ISR through both SA- and JA/ET-dependent signaling pathways [10,29].

Small RNAs (sRNAs) function to mediate plant defense responses against pathogens [8,30,31,32,33]. The sRNAs consist of small interfering RNAs (siRNAs) and microRNAs (miRNAs), which vary in biogenesis and precursor structure [8]; they can bind argonaute (AGO) proteins, forming a RNA-mediated silencing complex to regulate gene expression. Here, miR393 is the first example identified as PTI-related sRNA; its expression is elicited by flg22, a well-studied PAMP molecule, triggering PTI by inhibiting auxin signaling by silencing its receptors [34]. Moreover, miR773, miR160a, and miR398b act to regulate the deposition of callose, therefore participating in PTI [35]. On the other hand, some miRNAs are involved in ETI signaling; for example, miR393b* targets a Golgi-localized SNARE gene, which stimulates exocytosis of an antimicrobial pathogenesis-related protein, thus regulating plant defense [36]. The miRNAs also regulate the expression of defense-related host resistance (*R*) genes [37,38]. For instance, miR1885, whose expression is activated by Turnip mosaic virus (TuMV) infection in Brassica, targets the tollinterlekin receptor-nucleotidebinding site-leucinerich repeat (TIR-NBS-LRR) class resistance genes (*R* genes) [39]; miR482, whose expression is suppressed by virus infection, targets the NBS-LRR class *R* genes, and therefore suppresses tomato defense against pathogen attack [40]. In addition, enhanced expression of miR6019 and miR6020 leads to downregulation of *R* genes, causing attenuation of *R* gene-dependent defense responses to tobacco mosaic virus (TMV) in *Nicotiana benthamiana* [38]. Besides, miR472 downregulates PTI, as well as the ETI triggered by resistance to *Pseudomonas syringae* 5 (*RPS5*), by targeting coiled-coil (CC)-NBS-LRR genes in *Arabidopsis* [41].

We previously reported that AR156 triggers ISR to prevent *Pseudomonas syringae* pv. *tomato* (*Pst*)DC3000 and *B. cinerea* B1301 in *Arabidopsis* [10,28]. Moreover, we found that miR825 and miR825* in *Arabidopsis* act as negative regulators of AR156-mediated ISR to control *Pst* DC3000 by repressing the expression of defense-related genes [42]. On the basis of these findings, the present study was undertaken to elucidate the function of miR825 and miR825* in AR156-mediated ISR against *B. cinerea* B1301. As a result, Northern blotting revealed that upon challenge inoculation with *B. cinerea* B1301, stronger downregulation of miR825 and miR825^*^ expression occurred in AR156-pretreated plants than in nontreated control plants. On the other hand, miR825- and miR825*-overexpressing (OE) plants showed a higher susceptibility to *B. cinerea* B1301 than Col-0; in contrast, the short tandem target mimic (STTM) miR825 and miR825* (STTM825/825*) transgenic lines were more resistant to it. Moreover, upon *B. cinerea* B1301 infection, cellular defense responses (hydrogen peroxide production andcallose deposition) and expression of defense-related genes were stronger in AR156-pretreated plants from miR825/825* knockdown lines, but weaker in those from miR825 and miR825* OE plants than in Col-0 plants. We also identified a number of *R* genes of the TIR-NBS-LRR class as miR825* targets, which were expressed in a similar manner during AR156-triggered ISR. Furthermore, the target mutant plants were more prone to *B. cinerea* B1301 infection than Col-0; on the other hand, AR156 still induced an effective ISR in target mutant lines. This study indicated that miR825 and miR825* function to inhibit AR156-elicited ISR to control *B. cinerea* by repressing defense-related gene expression and cellular defense responses.

## 2. Results

### 2.1. miR825 and miR825* Expression was Suppressed in AR156-induced ISR to Prevent B. Cinerea in Arabidopsis

To decipher the function of miR825 and miR825* in AR156-triggered ISR to control *B. cinerea* B1301 in *Arabidopsis,* wild-type Col-0 plants were pretreated with AR156 or 0.85% NaCl as a root drench and then inoculated with *B. cinerea* B1301 or water (mock) at 7 days post-treatment (dpt). As shown in Figure 1A, at 2 days post-inoculation (dpi), nontreated control plants (control/mock) and AR156-treated plants (AR156/mock), both of which were mock-inoculated, showed no symptoms; in contrast, with *B. cinerea* B1301 inoculation, both nontreated control plants (control/*B. cinerea*) and AR156-treated plants (AR156/*B. cinerea*) developed typical symptoms of gray mold disease (severe necrotic lesions surrounding inoculating loci, leaf yellowing, and water-soaked spots covered with *B. cinerea* B1301 spores), while the symptoms were less pronounced in AR156/*B. cinerea* than incontrol/*B. cinerea*, indicating that AR156 pretreatment enhanced plant resistance to *B. cinerea* B1301. To investigate whether miR825 and miR825* function in the mediation of AR156-triggered ISR against *B. cinerea* B1301, we examined the expression of the two miRNAs in AR156/*B. cinerea*, control/*B. cinerea*, AR156/mock, and control/mock at 48 h post-inoculation (hpi). Northern blotting analysis confirmed that miR825* expression was downregulated with *B. cinerea* B1301infection in both AR156-treated and control plants. On the one hand, there was no appreciable difference in the level of miR825* expression between AR156/mock and control/mock, yet its level was significantly lower in AR156/*B. cinerea* than in control/*B. cinerea*, and the difference in the same parameter between control/mock and control/*B. cinerea* was smaller than that between AR156/mock and AR156/*B. cinerea* (Figure 1B). In addition, we also investigated the role of miR825 in ISR. There was no difference in miR825 expression observed between AR156/mock and control/mock, which was similar to the expression profile of miR825*, whereas miR825 expression was also downregulated upon *B. cinerea* B1301 infectionin the control plants, with the downregulation being significantly stronger in AR156-treated plants (Figure 1B). The results indicated that AR156 pretreatment resulted in significant suppression of the expression of miR825 and miR825*, especially during *B. cinerea* infection,suggesting that they might function as negative regulators of AR156-mediated ISR to prevent *B. cinerea* B1301.

### 2.2. miR825 and miR825* Participate in AR156-induced ISR

To further investigate whether miR825 and miR825* function in AR156-elicited ISR to control *B. cinerea* B1301, the ability of AR156 to trigger ISR was evaluated in the transgenic plants in which miR825 and miR825* wereknocked down by using short tandem target mimic (STTM) strategy, and in miR825 and miR825* OE plants. Two-way analysis of variance (ANOVA) with Tukey’s test was used to compare the levels of resistance to *B. cinerea* B1301 in AR156-treated and nontreated plants. Typical symptoms of gray mold disease (severe necrosis and water-soaked spots surrounded by the spores) appeared on leaves of control/*B. cinerea* from Col-0 at 2 dpi (Figure 2A); in comparison, disease symptoms were more severe in those from miR825/825* OE lines (#50 and #56), which formed larger necrotic lesions, but lighter in those from STTM825/825* transgenic lines (#1 and #3) (Figure 2A). However, AR156 pretreatment led to marked reductions in disease symptoms regardless of plant lines (Figure 2A). Compared to Col-0 plants, the diameter of leaf necrosis was smaller in the two STTM825/825* lines, but bigger in the two miR825/825* OE lines, and AR156-treated plants from STTM825/825* lines developed significantly (** *P* < 0.01) smaller necrotic lesions on leaves than the corresponding control plants (Figure 2B). On the other hand, compared with Col-0, STTM825/825* lines (#1 and #3) expressed stronger resistance to *Phytophthora capsici* (*P. capsici*) (Appendix A) and cucumber mosaic virus (CMV) (Appendix A), and showed significant (** *P* < 0.01) reductions in the lesion diameter on the leaf infected by *P. capsici* (Appendix A) and in that by CMV (Appendix A), indicating that miR825 and miR825* act as negative regulators of *Arabidopsis* resistance to various pathogens (*B. cinerea*, *P. capsici*, and CMV). Additionally, qRT-PCR was conducted to measure *B. cinerea* B1301 biomass at 2 dpi in infected leaves. Coincident with the reduction in disease symptoms on AR156-treated leaves, the biomass of *B. cinerea* B1301 was consistently lower in leaf tissues of AR156-pretreated plants with three different genotypes than in respective control plants; with AR156 pretreatment, the fungal biomass was lower in the STTM825/825* lines, but higher in the miR825/825* OE lines than in Col-0 plants (Figure 2C). The two-way ANOVA revealed that the interaction between treatment (AR156, control) and genotype was statistically significant, and that AR156-pretreated plants, with three different genotypes overall, showed significantly higher levels of resistance to *B. cinerea* B1301 than respective control plants. Collectively, these results implicate that miR825 and miR825* function to suppress innate immunity in *Arabidopsis* that protect plants from the infection of a wide variety of pathogens (*B. cinerea*, *P. capsici*, and CMV).

### 2.3. miR825 and miR825* Affect Defense-related Gene Expression and Cellular Defense Responses in AR156-primed Plants upon Pathogen Attack

To identify the signaling pathway(s) involved in AR156-induced ISR, qRT-PCR was used to examine expression levels of the response genes of the SA-signaling pathway (*PR1*, *PR2*, and *PR5*) and that of the JA/ET-signaling pathway (*PDF1*.2). The expression of these genes in plants with different genotypes was analyzed with two-way ANOVA. Without *B. cinerea* B1301infection, their expression levels in the miR825/825* OE lines (#50 and #56) and STTM825/825* transgenic lines (#1 and #3) were comparable to those in the wild-type plants at 48 hpi (Figure 3A). However, their transcription was significantly enhanced in AR156/*B. cinerea* relative to that in control/*B. cinerea* at 48 hpi, regardless of plant lines (Figure 3A). Moreover, the two-way ANOVA revealed that the interaction between treatment effect and genotype effect was statistically significant; with AR156 pretreatment, the transcriptional levels of these genes significantly increased in the STTM825/825* transgenic plants, but declined in the miR825/825* OE plants compared with those in Col-0 plants (Figure 3A), which conformed to the function of miR825/825* as negative regulators of AR156-mediated ISR to prevent *B. cinerea*.

We also investigated whether miR825 and miR825* act in AR156 priming of the plant for potentiated cellular defense responses—represented by the accumulation of hydrogen peroxide and the deposition of callose—in *Arabidopsis* upon pathogen attack. As shown in Figure 3B, the two events were evident at 12 hpi in AR156/*B. cinerea* of both Col-0 and STTM825/825* transgenic plants (lines 1 and 3), but much weaker in that of miR825/825^*^ OE plants (lines 50 and 56); meanwhile, they were absent in control/*B. cinerea*. With AR156 pretreatment, these defense responses were potentiated in Col-0 and the two STTM825/825* transgenic lines at 24 hpi; meanwhile, they were weaker in both control/*B. cinerea* and AR156/*B. cinerea* of miR825/825* OE plants (Figure 3B). The observed results indicated that upon pathogen attack, AR156-treated *Arabidopsis* plants expressed faster and stronger cellular defense responses, which, however, could be dampened by miR825 and miR825*.

### 2.4. miR825 and miR825* Participate in Regulation of PTI Components in AR156-elicited ISR

With the purpose of elucidating the role of miR825 and miR825* in AR156-triggered ISR and PTI, the phosphorylation of MPK3 and MPK6 was examined in both control and AR156-pretreated plants inoculated with *B. cinerea* B1301. Using antibodies specifically recognizing MPK3 and MPK6, their phosphorylation was detected in leaves of control/*B. cinerea* of Col-0 at 10 min post-inoculation (mpi); then, their intensity gradually declined at 30 and 60 mpi. However, AR156 pretreatment caused sustained phosphorylation of the two kinases in leaves of Col-0 plants. The intensity increased with time, peaking at 60 mpi (Figure 4A). In control/*B. cinerea* and AR156/*B. cinerea* from miR825/825* OE line (#50), the tendency of MPK3 and MPK6 phosphorylation was the same as that in the corresponding treatments from Col-0 (Figure 4B). Notably, the phosphorylation of the two kinases sustained from 10 to 60 mpi in both control/*B. cinerea* and AR156/*B. cinerea* of miR825/825* knockdown line (#1), with the intensityremaining substantially higher in AR156/*B. cinerea* than in control/*B. cinerea* throughout the whole time course (Figure 4C). These results suggested that AR156-triggered ISR involves an enhanced and sustained PTI response, in which miR825 and miR825* play a negative regulatory role.

To further test the hypothesis, we evaluated the transcription levels of two PTI marker genes: the flg22*-*induced receptor-like kinase 1 (*FRK1*) and *WRKY53* [43,44]. Differences in their expression between AR156-pretreated and control plants with different genotypes upon *B. cinerea* B1301 infection were analyzed by two-way ANOVA. The qRT-PCR assay revealed that the expression of *FRK1* and *WRKY53* was enhanced and sustained in control/*B. cinerea* of Col-0, miR825/825* OE line (#50), and STTM825/825* transgenic line (#1) from 10 to 60 mpi. However, their expression significantly increased at 30 and 60 mpi in the AR156/*B. cinerea* of STTM825/825* lines relative to that in the corresponding control plants (Figure 4D). The two-way ANOVA revealed a significant (*P* < 0.01) interactive effect of treatment and genotype. These results confirmed that miR825 and miR825* negatively regulate the expression of PTI components in AR156-mediated ISR.

### 2.5. Target Genes of miR825 and miR825* are Expressed in a Similar Manner in AR156-induced ISR

To investigate whether miR825 and miR825* modulate ISR by affecting the expression of their targets genes, qRT-PCR was conducted to examine expression patterns of miR825/825* target genes in control and AR156-pretreated plants of Col-0, miR825/825* OE lines (#50 and #56), and STTM825/825* lines (#1 and #3), with or without *B. cinerea* B1301 inoculation. The expression of the targeted genes in plants with different genotypes treated with or without AR156 was analyzed using two-way ANOVA. As expected, the results of qRT-PCR indicated that the transcriptional levels of three miR825*-targeted genes (*AT5G40910*, *AT5G38850*, and *AT3G04220*) and a miR825-targeted gene (*AT5G44940*) were all significantly improved in leaves of both control and AR156-pretreated plants from miR825/825* knockdown lines (#1 and #3), with or without *B. cinerea* B1301 inoculation, compared with those from Col-0 at 48 hpi. In contrast, their expression levels were consistently reduced in four treatments (control/mock, AR156/mock, control/*B. cinerea*, and AR156/*B. cinerea*) from miR825/825* OE lines (#50 and #56) compared with Col-0 at the same time. Moreover, AR156-induced transcription of the four target genes was significantly stimulated in the STTM825/825* transgenic plants, but attenuated in the miR825/825* OE plants relative to that in Col-0 plants (Figure 5A). At the same time, AR156 pretreatment coupled with *B. cinerea* B1301 infection led to conspicuous up- and downregulation of these genes in STTM825/825* and miR825/825* OE plants, respectively, compared with that in Col-0 plants (Figure 5A). Two-way ANOVA revealed that the interaction between treatment effect and genotype effect was statistically significant. Hence, it is clear that miR825 and mi825* functionto suppress AR156-elicited ISR.

To further confirm that these genes are targeted by miR825 and miR825*, the two miRNAs were separately transiently co-expressed with a target gene fused with the HA-tag in *Nicotiana benthamiana*. An anti-HA antibody was used to detect the proteins encoded by *AT5G40910, AT5G38850, AT3G04220,* and *AT5G44940.* As a result,their expressionwas apparently repressed by the co-expression of miR825/825*, but not by miR319b, which is irrelevant to these genes (Figure 5B). Taken together, these results signified that the four genes are authentic miR825- or miR825*-targeted genes participating in AR156-induced ISR against *B. cinerea* B1301.

### 2.6. Plants Silencing miR825/miR825*-targeted Genes are More **S**usceptible to B. cinerea

To clarify the function of the target genes of miR825 and miR825*, they were evaluated in a functional analysis by using the null mutants *at5g40910* (SALK_043422C), *at5g38850* (SALK_134889C), *at3g04220* (CS384498), and *at5g44940* (SALK_021558C), each with a T-DNA insertion in an intron or exon (Appendix A). Semi-quantitative RT-PCR confirmed that the two mutants *At5g38850* and *At3g04220* were homozygous mutants (Appendix A); however, this assay also indicated that we failed to obtain the homozygous mutants of *AT5G40910* and *AT5G44940* (Appendix A), and we speculate that this failure might be caused by homozygous infertility or T-DNA insertion mutation. The germination rate for seeds was similar among Col-0, *at5g38850*, and *at3g04220* (Appendix A). The mutants *at5g38850* and *at3g04220* phenotypically developed to a lesser degree than Col-0 plants (Appendix A). To elucidate the role of these target genes in AR156-induced ISR to control *B. cinerea* B1301 in *Arabidopsis*, we assessed disease development in Col-0 and the two mutant lines at 2 dpi. Two-way ANOVA with Tukey’s test was conducted to compare the differences in disease severity between AR156-treated and nontreated plants, with different genotypes infected with the pathogen. As shown in Figure 6a, *B. cinerea* B1301 infection caused severe disease symptoms on leaves of control plants from both Col-0 and the two mutant lines, while the symptoms were more severe in the mutants than in Col-0, indicating the mutant lines were more prone to the attack by *B. cinerea* B1301 than Col-0 (Figure 6A). However, AR156 pretreatment caused a noticeable attenuation of disease symptoms on plants of Col-0 and the two mutants, demonstrating that AR156 induced effective ISR in both Col-0 and the mutants. Consistently, AR156-treated plants of Col-0 and target mutant lines developed significantly (** *P* < 0.01) smaller necrotic lesions on leaves than their respective control plants; on the other hand, the lesions in the target mutants were larger than those in Col-0 plants (Figure 6B). The fungal biomass in plants of Col-0 and the two target mutant lines were also determined by qRT-PCR. Consistent with the inhibitory effect of AR156 pretreatment on the disease symptoms observed on the leaf surface, the growth of *B. cinerea* B1301 at 2 dpi was significantly slower in AR156/*B. cinerea* than in control/*B. cinerea*, regardless of plant lines; on the other hand, the fungal biomass increased in target mutants more than in Col-0 plants, regardless of pretreatment (AR156 or control) (Figure 6C). Importantly, the two-way ANOVA showed a significant interactive effect of treatment and genotype, as well as a significant treatment effect. This was demonstrated by the overall levels of resistance to *B. cinerea* B1301 in AR156-pretreated plants significantly exceeding those in control plants across different genotypes. Taken together, the target mutant plants showed increased susceptibility to the pathogen, yet AR156 still retained the capacity to trigger ISR in them.

## 3. Discussion

Biological control is an effective strategy for protecting plants from pathogen infection. Our earlier researches demonstrated that the PGPR strain *B. cereus* AR156 can elicit ISR against *Pst* DC3000 in *Arabidopsis* through activating both the SA-dependentand the JA/ET-dependent signaling pathway [10,45]. More recently, we found that AR156 also triggers ISR effective in controlling *B. cinerea* B1301, a necrotrophic pathogen, which is dependent on the JA/ET pathway, but not the SA pathway [28]. Moreover, documented studies have elucidated the molecular processes and the components of signaling pathways contributing to the interactions between host plants and ISR elicitors [12,18]. We also reported that miR825 and miR825* downregulate AR156-meditated ISR against *Pst* [42]. On the other hand, it is known that miRNAs also function to suppress plant innate immunity. Without pathogen infection, they inhibit plant defense responses; however, under pathogenic stress, healthy plants express a variety of immune responses by relieving miRNA-regulated repression [32,42,46]. The present study showed that AR156 pretreatment triggers ISR to prevent *B. cinerea* B1301 by inhibiting miR825 and miR825* expression, thereby inducing the expression of their targeted defense genes in *Arabidopsis*.

It is known that defense-related genes are induced when plants and pathogens interact under primed conditions. *PR1* gene expression was stronger in AR156/*Pst* samples than in control/*Pst* [10,20]. The induced defense-related gene expression has also been observed in ISR to some necrotrophic and hemibiotrophic pathogens; for example, the expression of defense genes was induced in rice plants primed by *Harpophoraoryzae* and infected by *Magnaportheoryzae* [47], in tomato plants primed by *Bacillus subtilis* CBR05 and infected by *Erwinia carotovora* subsp. *Carotovora* [48], and in *Arabidopsis* plants pretreated by *B. cereus* AR156 and infected by *B. cinerea* B1301 [28]. In line with this, it has been documented that enhanced *PR* gene expression promotes defense responses to necrotrophic and hemibiotrophic pathogens [47,48]. In this study, the transcriptional levels of *PR1*, *PR2*, *PR5*, and *PDF1.2* in plants with different genotypes were analyzed by two-way ANOVA. On the one hand, AR156 treatment significantly stimulated their transcription and enhanced hydrogen peroxide accumulation and callosedeposition in Col-0, STTM825/825*, and miR825/825* OE plants upon infection with *B. cinerea* B1301. On the other hand, these events were significantly stronger in the STTM825/825* transgenic plants, but weaker in the miR825/825* OE plants than those in Col-0 plants (Figure 3). Importantly, the interaction between treatment effect and genotype effect was statistically very significant (*P* < 0.01) (Figure 3A). Our results indicated that miR825 and miR825* played a negative regulatory role in AR156-elicited potentiation of cellular defense responses. This finding is supported by prior studies demonstrating that potentiated accumulation of hydrogen peroxide and deposition of callose are important components of plant immune responses to the attack by *B. cinerea* B1301 [49,50].

PTI is indispensable to plant defense against *B. cinerea* B1301. The earliest events in PTI include activation of mitogen-activated protein kinases (MAPKs) [44,51,52]. MAPK cascade signaling is essential for plant defense against pathogen infections. In the current research, MPK3 and MPK6 phosphorylation was detected at 10 mpi in both control/*B. cinerea* and AR156/*B. cinerea* from plants of Col-0, miR825/825* OE line (#50), and miR825/825* knockdown line (#1); at 10 to 60 mpi, the phosphorylation was considerably stronger in AR156/*B. cinerea* than in control/*B. cinerea*, regardless of plant line (Figure 4A). Therefore, the treatment with AR156 enhanced MPK3 and MPK6 phosphorylation. Many studies have shown that MPK3 and MPK6 are positive mediators of plant disease resistance [52,53,54], being critical for plant immunity to *B. cinerea* B1301 [55,56,57]. In the present study, the expression of *FRK1* and *WRKY53,* two MAMP-specific early-defense marker genes, was down- and upregulated in miR825/825* OE and knockdown lines, respectively, compared with that in Col-0 plants, which demonstrated the involvement of miR825 and miR825* in innate immunity. These results also consolidated our conclusion that they are negative regulators of AR156-induced ISR to control *B. cinerea* B1301. Nonetheless, the exact function of miR825 and miR825* in sub-branches of PTI remains to be deciphered. Given that miR825 and miR825* also negatively regulate defense responses in AR156-induced ISR to *Pst* DC3000 [42], they might regulate plant resistance to the bacterium and the fungus by using similar mechanisms.

An *R* gene family of the TIR-NBS-LRR class is targeted by miR825*, whose target site lies in the region that encodes the TIR domain, which is commonly shared by the proteins of the TIR-NBS-LRR family [42]; miR825* may target a number of *R* genes [42].On the other hand, miR825 targets AT5G44940, a member of the F-box/RNI-like superfamily. Moreover, proteins in the F-box/RNI-like superfamily are involved in protein ubiquitination; transport inhibitor response 1 (TIR1), a F-box auxin receptor, modulates auxin signaling by triggering auxin/indole-3-acetic acid (Aux/IAA) proteasomal degradation [58]. *TIR1* is targeted by miR393, whose expression is activated by flg22, thus contributing to PTI [34]. Our study demonstrated that miR825 and miR825* act as negative regulators of AR156-triggered ISR to prevent *B. cinerea* B1301 by repressing the transcription of the targeted defense genes activated by AR156 in plants upon its attack.In some cases, correlation between resistance and induction of *R* genes, or reduction of miRNA and miRNA-triggered phasiRNA, was observed among different crop varieties [59,60,61]. In *Arabidopsis*, the miR472-RDR6 pathway was shown to negatively regulate both PTI and ETI constitutively and inhibit induction of *R* genes during infection [41]. Besides biotic stress, various abiotic stresses are also shown to modulate miRNA-mediated *R* gene regulation. It is reported that in *Populus trichocarpa*, miR482 accumulation in developing xylem was decreased by tension and compression stress [62]. The miRNA-mediated regulation of plant *R* genes has emerged as an important general mechanism operating in all R gene-encoding plants examined to date [63]. Additionally, the level of AR156-mediated ISR tocontrol *B. cinerea* B1301 was lower in each of the target mutants than in wild type plants (Figure 6). The two-way ANOVA demonstrated that the interaction between treatment effect and genotype effect was statistically significant, and that the levels of resistance to *B. cinerea* B1301 in AR156-treated plants overall significantly surpassed those in control plants across different genotypes. Hence, these target genes may participate in AR156-induced ISR.

The present study showed that miR825 and miR825* suppress AR156-elicited ISR to prevent *B. cinerea* B1301 stress, as demonstrated by the down- and upregulation of several defense-related genes in 825/825* OE andSTTM825/825* plants, respectively, compared with Col-0 plants. Moreover, STTM825/825* plants without AR156 treatment showed stronger resistance than Col-0 to a wide variety of pathogens, including viruses and oomycetes, which indicated that miR825 and miR825* also function to negatively regulate innate immunity. Therefore, the findings of this study have significant implications for effective application of the two miRNAs for plant protection.

## 4. Materials and Methods

### 4.1. Plant Materials and Growth Conditions

Plants of *Arabidopsisthaliana* and *Nicotiana benthamiana* were cultivated in a growth chamber maintained at 23 ± 1 °C in a photoperiod of 12 h light/12 h dark. *Arabidopsis* lines used in this study were as follows: Col-0 (*Arabidopsisthaliana* ecotype); two transgenic plant lines (#50 and #56), each overexpressing both miR825 and miR825*; two STTM825/825* transgenic plant lines (#1 and #3) [42]; and the T-DNA insertion mutants *at5g44940* (SALK_021558C), *at5g38850* (SALK_134889C), *at5g40910* (SALK_043422C), and *at3g04220* (CS384498). The seeds of Col-0 and *Nicotiana benthamiana* were kindly provided by Hansong Dong (Nanjing Agricultural University, Nanjing, China). Dongdong Niu constructed and identified overexpression and STTM of miR825/825* transgenic plant lines, and the seeds of four T-DNA insertion mutants were obtained from the College of Arts and Sciences Arabidopsis Biological Resource Center (ABRC, The Ohio State University, Columbus, OH, USA). Homozygous mutant plants were identified by using semi-quantitative RT-PCR, and the primers are shown in Appendix A. Three-week-old plants were used in all experiments.

### 4.2. Induction Treatments and Pathogen Infection

Thetested PGPR strain *B. cereus* AR156 was cultured on Luria–Bertani (LB) agar plates, which were incubated at 28 °C for 24 h. Bacteria were subsequently collected by centrifugation, followed by their resuspension in a sterile 0.85% NaCl solution, with a final concentration of 5 × 10^8^ CFU/mL. To prepare the inoculum of *B. cinerea* B1301, the strain was cultured on potato sucrose agar (PSA) for 7 days; its spores were collected in H_2_O, from which a spore suspension with a final concentration of 1 × 10^6^ spores/mL was prepared. Each plant was applied with 7 mL of the AR156 cell suspension or a sterile 0.85% NaCl solution (the control) as a root drench. Seven days later, plants were challenge-inoculated by dropping a 10 μL droplet of the spore suspension of *B. cinerea* B1301 on the midvein of each side of the leaves. Then, all plants were grown in a growth chamber maintained at 23 ± 1 °C and with 70% relative humidity for 2 days. Transcript levels of *B. cinerea* actin gene (*BcActin*) were measured to determine in planta biomassof *B. cinerea* B1301, employing the *Arabidopsis* actin 1 gene (*AtActin1*) as an internal control. The ratio of *BcActin* to *AtActin1* (*BcActin/AtActin1*) was calculated to indicate relative fungalbiomass.

For *Phytophthora capsici* (*P. capsici*) infection of *Arabidopsis*, *P. capsici* was cultured as reported by Lu [64]. The number of *P. capsici* zoospores in 10 μL water was counted under a microscope (Leica, Heidelberg, Germany) to determine its concentration. Leaves were detached from plants of Col-0, miR825/825* OE lines (#50 and #56), and STTM825/825* lines (#1 and #3), and then each leaf was put on a piece of wet paper in a petri dish. Approximately 100 *P. capsici* zoospores were dropped onto the center of each leaf, which were then incubated in a growth room maintained at 25 °C in darkness.

For virus inoculations, transient expression assays in *N. benthamiana* were conducted by syringe-infiltrating leaves of three-week-old *Nicotiana benthamiana* plants with a cell suspension of *Agrobacterium* (OD_600_ = 1.0) harboring constructs containing cucumber mosaic virus (CMV). *N. benthamiana* leaves were sampled at 7 days post-inoculation (dpi); ground in 1× PBS; and then rub-inoculated onto leaves of plants from Col-0, miR825/825* OE lines (#50 and #56), and STTM825/825* lines (#1 and #3). All virus-inoculated plants were grown in growth chambers (model GXZ500D, Jiangnan Motor Factory, Ningbo, China) maintained at 25 °C with 8 h light/16 h dark.

### 4.3. RNA Extraction, Quantitative RT-PCR, and RNA Gel Blot Analysis

Total RNA in *Arabidopsis* leaves was extracted with TRIzol reagent (Invitrogen, San Diego, CA, USA). Briefly, cDNA was synthesized by using 1 μg total RNA with a commercial reverse transcription system (TaKaRa Biotech, Dalian, China). Real-time quantitative RT-PCR was performed by using 2 μL cDNA that was 10× dilution of the synthesis product, under the following reaction conditions: 40 cycles of denaturing at 95 °C for 30 s, annealing at 55 °C for 30 s, and extension at 72 °C for 34 s. Each sample comprised three replications. The data were normalized with *AtActin1*.

RNA gel blot analysis was conducted using the method of Yan [65]. RNA samples were resolved on a denaturing 17% polyacrylamide gel; subsequently, the RNA molecules separated by the electrophoresis were transferred to Hybond-N^+^ membranes electrophoretically (Amersham BioScience, Singapore). The sequences of primers used in the quantitative RT-PCR and Northern blotting assays are listed in Appendix A.

### 4.4. Detection of Hydrogen Peroxide Accumulation and Callose Deposition

Three-week-old *Arabidopsis* plants from Col-0, miR825/825* OE lines (#50 and #56), and STTM825/825* lines (#1 and #3) were pretreated by applying 7 mL of the AR156 suspension at 5 × 10^8^ CFU/mL or a sterile 0.85% NaCl solution (the control) to each plant as a root drench. At 7 dpt, *B. cinerea* B1301 spores were collected to make a spore suspension with the final concentration at 1 × 10^6^ spores/mL. Plants were challenge-inoculated by dropping a 10 μL droplet of the spore suspension on the midvein of each side of a leaf. At 12 and 24 h post-inoculation (hpi), *Arabidopsis* leaves were sampled and then processed to detect the accumulation of hydrogen peroxide and the deposition of callose using the protocols described in our earlier study [10]. In brief, for ROS accumulation, leaves of each sample were collected from at least three different plants, which were stained with a diaminobenzidine (DAB) solution (1 mg/mL, pH 3.8) for 8 h in darkness at 25–28 °C. Subsequently, 96% (*v/v*) ethanol was used to clear the stained leaves, which were finally preserved in 50% (*v/v*) ethanol. Under a light microscope, hydrogen peroxide was visualized as dark-brown precipitate in a DAB-stained leaf.

To detect the deposition of callose, leaves were soaked in 5 mL of a destaining solution consisting of acetic acid and ethanol in a ratio of 5:95 (*v/v*), followed by vacuum infiltration for 5–10 min. Leaf chlorophyll was cleared by incubating leaves in a 60 °C water bath for 20–30 min. After being gently rinsed with water, the resultant chlorophyll-free leaves were immersed in 3–5 mL of a staining solution that contained 0.01% (*w/v*) aniline blue and 150 mM K_2_HPO_4_ (pH 9.5) in the darkness for 2–4 h. After being gently rinsed with water, the stained leaves were mounted on microscope slides for observation under an epifluorescence microscope equipped with a UV excitation filter.

### 4.5. Protein Extraction and Western Blotting Analysis

Three-week-old plants of *Arabidopsis* Col-0, miR825/825* OE transgenic line #50, and STTM825/825* transgenic line #1 were applied with AR156 or 0.85% NaCl (the control), as described above. At 7 dpt, challenge inoculation was carried out. Leaves were sampled at 0, 10, 30, and 60 mpi. Subsequently, they were ground in liquid nitrogen, followed by extraction of total proteins with a 2× SDS loading buffer. The extracted proteins were separated on 12% SDS-PAGE gels, and then the separated protein molecules were transferred onto nitrocellulose membranes, followed by blotting with a monoclonal mouse anti-HA (1:5000 dilution) and a monoclonal mouse anti-α tubulin (1:5,000 dilution). To measure MAPK activity, samples were subjected to Western blotting with monoclonal rabbit phospho-p44/42 MAPK (Erk1/2) (Thr202/Tyr204) (D13.14.4E) XP antibodies (Cell Signaling Technology, Danvers, MA, USA, #4370S, 1:2000 dilution), with α-tubulin included as a loading control.

### 4.6. Transient Expression Analysis in Nicotiana benthamiana

To generate the expression constructs of miR825, miR825*, and miR319b, the primary constructs amiR825, amiR825* and amiR319b were engineered from pRS300 (miR825, miR825*, and miR319b) *Arabidopsis thaliana* by using the WMD3 online tool (http://wmd3.weigelworld.org). These were subsequently cloned into a pEarleyGate (*pEG202*) destination vector by LR clonase II (Invitrogen, Carlsbad, CA, USA). The corresponding miRNA overexpression plant lines were generated by cloning the coding sequences of *At5G40910*, *At5G38850*, *At3G04220*, and *At5G44940* into the *pEG201* vector.

In transient co-expression and co-immunoprecipitation assays, leaves of three-week-old *N. benthamiana* plants were syringe-infiltrated with a cell suspension of *Agrobacterium tumefaciens* GV3101 (OD_600_ = 1.0) harboring constructs containing miR825 (*pEG202*) and miR825* (*pEG202*), as well as that of *A. tumefaciens* GV3101 (OD_600_ = 1.0) containing the coding sequences of the four target genes (*pEG201*), while miR319b (*pEG202*) was employed as the control. Leaf tissues were collected at 48 hpi and processed as described above, and protein expression was detected by Western blotting.

### 4.7. Statistical Analysis

Two-way ANOVA analysis with Tukey’s test was conducted to detect the interaction between two independent variables: treatment (AR156 or control) and plant genotype. The Student’s *t*-test was carried out to assess statistical significance for all pairwise comparisons. Single and double asterisks above the columns in figures indicate significant (* *P* < 0.05) and very significant (** *P* < 0.01) differences, respectively, between AR156 and control treatments in plants with different genotypes, as well as those between different genotypes. Standard deviations were calculated. All analyses were conducted with SPSS software for Windows version 19.0 (IBM Co., Armonk, NY, USA).

## Figures and Tables

**Figure 1 ijms-20-05032-f001:**
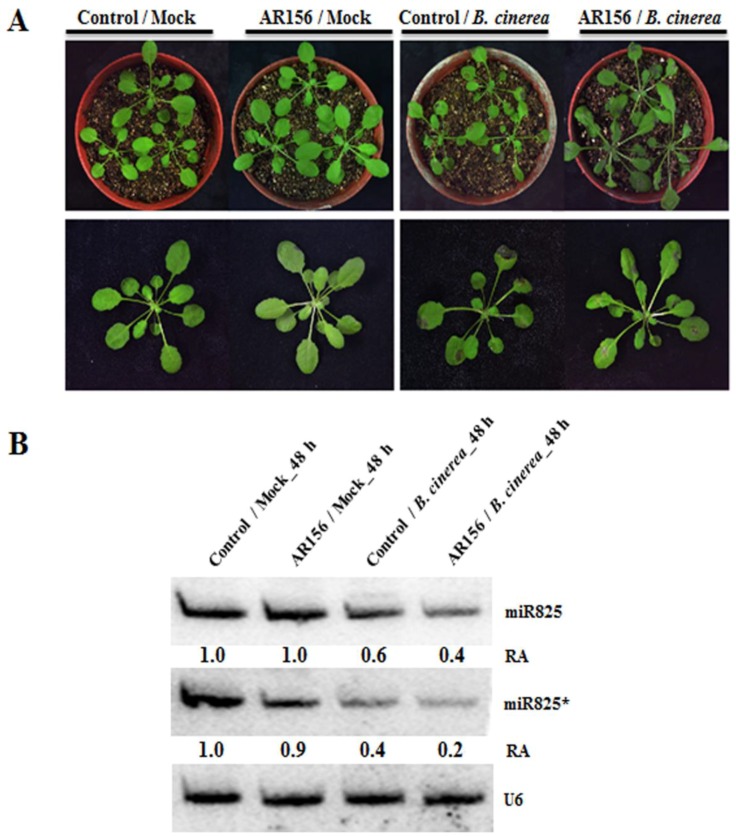
Theexpression level of miR825 and miR825* in *Arabidopsis*. Three-week-old *Arabidopsis* Col-0 plants were pretreated with AR156 at 5 × 10^8^ CFU/mL or 0.85% NaCl (the control) as a root drench; at 7 days post-treatment (dpt), challenge inoculation was carried out by dropping 10 µl of a *B. cinerea* B1301 spore suspension at 1 × 10^6^ spores/mL or H_2_O (mock) on the midvein of each side of a leaf. (**A**) At 48 h post-inoculation (hpi), disease symptoms on leaves were observed and photos were taken. (**B**) The miR825 and miR825* expression in different treatments was detected by Northern blotting. Total RNA was extracted from leaves of *Arabidopsis* Col-0 plants in four treatments (control/mock, AR156/mock, control/*B. cinerea*, and AR156/*B. cinerea*) at 48 hpi. RNA blots were probed with DNA oligonucleotides complementary to miR825 or miR825*. U6 served as a loading control. Relative abundance (RA) levels are indicated. The experiments were repeated three times, which yielded similar results.

**Figure 2 ijms-20-05032-f002:**
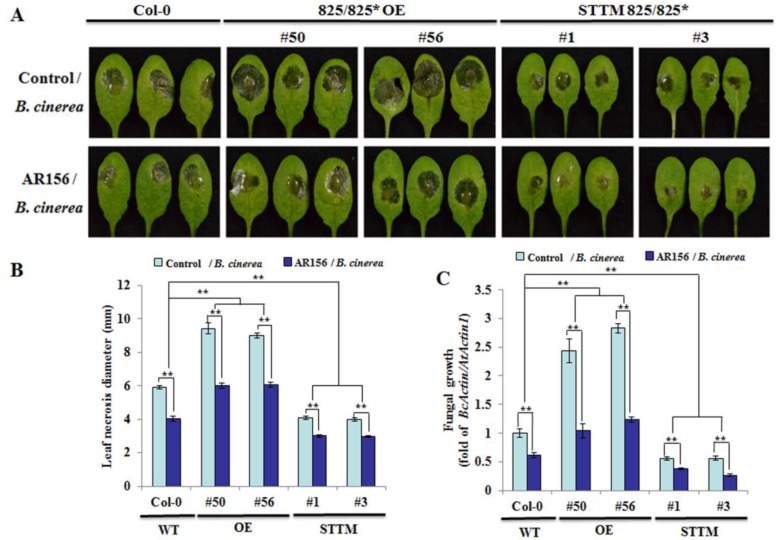
The levels of resistance to *B. cinerea* B1301 with different genotypes.Three-week-old *Arabidopsis* plants from Col-0, miR825/825* overexpressing (OE) lines (#50 and #56), and STTM825/825* lines (#1 and #3) were pretreated with AR156 at 5 × 10^8^ CFU/mL or 0.85% NaCl (control) as a root drench; at 7 dpt, all plants were infected with *B. cinerea* B1301 by dropping 10 μL of its spore suspension (1 × 10^6^ spores/mL) on the midvein of each side of a leaf. (**A**) Disease symptoms on leaves of each tested plant line at 2 dpi. (**B**) Necrotic lesions were evaluated at 2 dpi by determining their average diameter on one leaf per plant, with a total of three plants evaluated for each sample. (**C**) In planta fungal growth in leaves of each tested plant line. The biomass of *B. cinerea* B1301 was determined by simultaneously quantifying the transcripts of *B. cinerea actin* gene (*BcActin*) and the *Arabidopsis* actin 1gene (*AtActin1*). Relative fungal biomass was indicated by *BcActin/AtActin1,* the ratio of *BcActin* to *AtActin1*. Two-way analysis of variance (ANOVA) showed significant effects of genotype (*P* < 0.01) and treatment (*P* < 0.01), and a significant interactive effect (*P* < 0.01); ** *P* < 0.01. The data are presented as mean ± SD from three biological replicates. The experiments wererepeated three times, bringing similar results.

**Figure 3 ijms-20-05032-f003:**
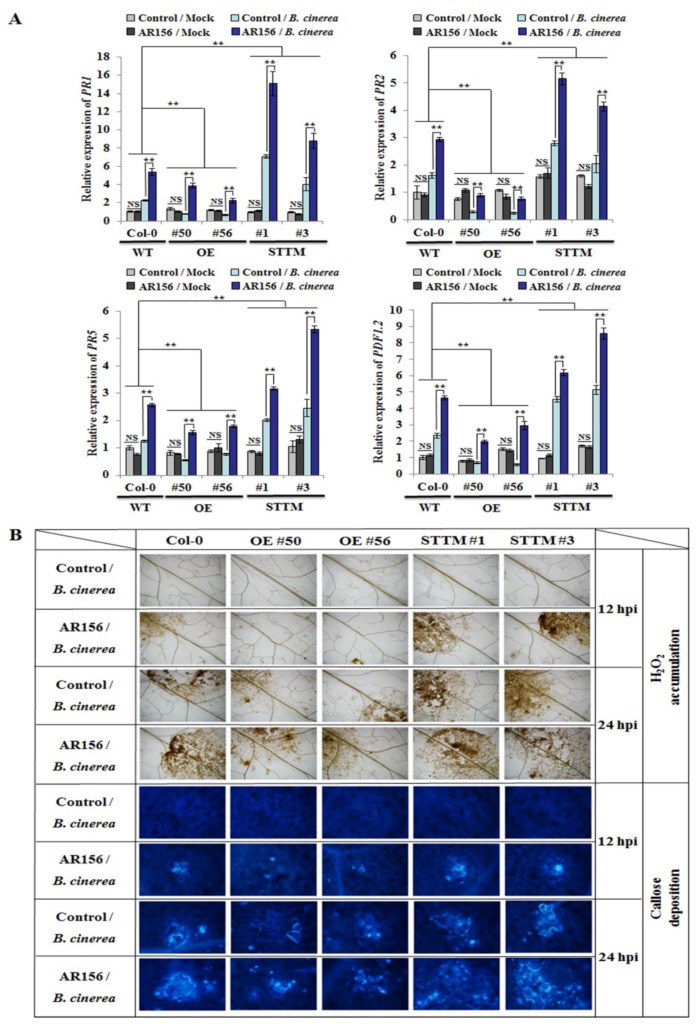
The defense gene expression and cellular defense responses in different genotypes.Roots of three-week-old plants from Col-0, miR825/825* OE lines (#50 and #56), and STTM825/825* lines (#1 and #3) were drenched with AR156 at 5 × 10^8^ CFU/mL or 0.85% NaCl (control). After 7 days, the leaves of these plants were inoculated with a *B. cinerea* B1301 spore suspension (1 × 10^6^ spores/mL) or H_2_O (mock), which were collected at 12, 24, and 48 hpi. (**A**) Transcription of defense-related genes upon attack by *B. cinerea* B1301. Using qRT-PCR, transcriptional levels of defense genes in leaves collected at 48 hpi were examined and normalized with those of *AtActin1*. The data are presented as mean ± SD from three biological replicates. The assay was repeated three times, yielding similar results. Two-way ANOVA showed significant effects of genotype on the expression of *PR1*, *PR2, PR5*, or *PDF1.2* (*P* < 0.01 for the differences between WT and OE, and also for those between WT and STTM); and a significant treatment effect (*P* < 0.01) and a significant interactive effect (*P* < 0.01) on the expression of each tested gene. Note: NS, not significant; ** *P* < 0.01. (**B**) Hydrogen peroxide accumulation and callose deposition in situ detected at 12 and 24 hpi in leaves inoculated with *B. cinerea* B1301. Accumulation of H_2_O_2_ was examined by diaminobenzidine (DAB) staining; with aniline blue staining, callose deposition was visible under light and epifluorescence microscopes attached to a UV excitation filter.

**Figure 4 ijms-20-05032-f004:**
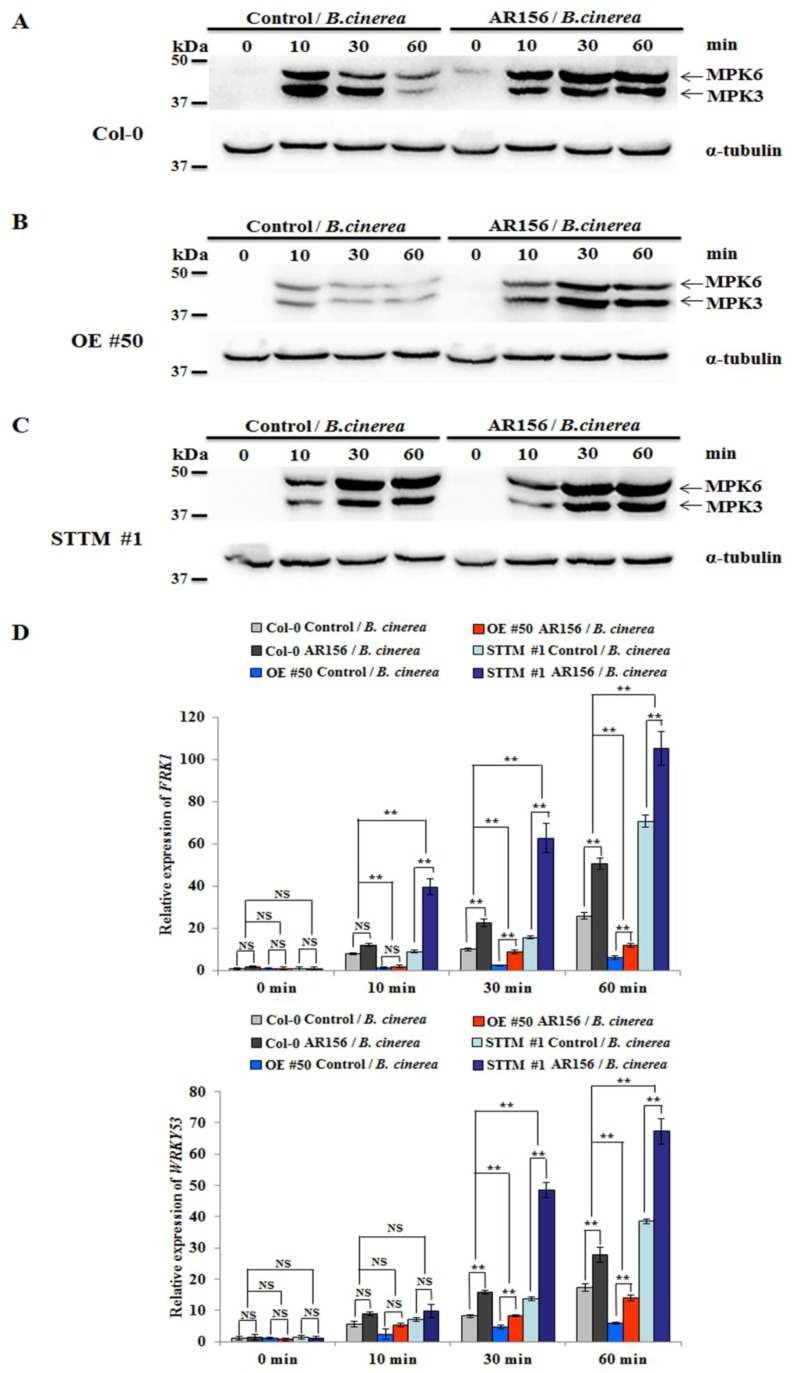
The role of miR825 and miR825* in AR156-triggered ISR and PTI. Three-week-old *Arabidopsis* plants from Col-0, miR825/825* OE line (#50), and STTM825/825* transgenic lines (#1) were pretreated with AR156 at 5 × 10^8^ CFU/mL or 0.85% NaCl (the control) as a root drench. At 7 dpt, they were challenge-inoculated with *B. cinerea* B1301 as depicted above at 0, 10, 30, and 60 mpi.(**A**–**C**) Leaves of each tested plant line were collected at 0, 10, 30, and 60 mpi; and phosphorylated MPK3 and MPK6 were detected by Western blotting, using α-tubulin as an equal loading control. (**D**)The transcriptional levels of *FRK1* and *WRKY53* in Col-0, miR825/825* OE line (#50), and STTM825/825* transgenic line (#1) were evaluated by means of real-time RT-PCR. Leaves were sampled at 0, 10, 30 and 60 mpi. *AtActin1* mRNA was included as an internal control. Two-way ANOVA showed significant effects of genotype on the expression of *FRK1* and *WRKY53* (*P* < 0.01 for the differences between WT and OE, and also for those between WT and STTM); and a significant treatment effect (*P* < 0.01) and a significant interactive effect (*P* < 0.01) on the expression of each tested gene. Note: NS, not significant; ** *P* < 0.01. The data are presented as mean ± SD from three biological replicates. The assays were repeated three times, giving similar results.

**Figure 5 ijms-20-05032-f005:**
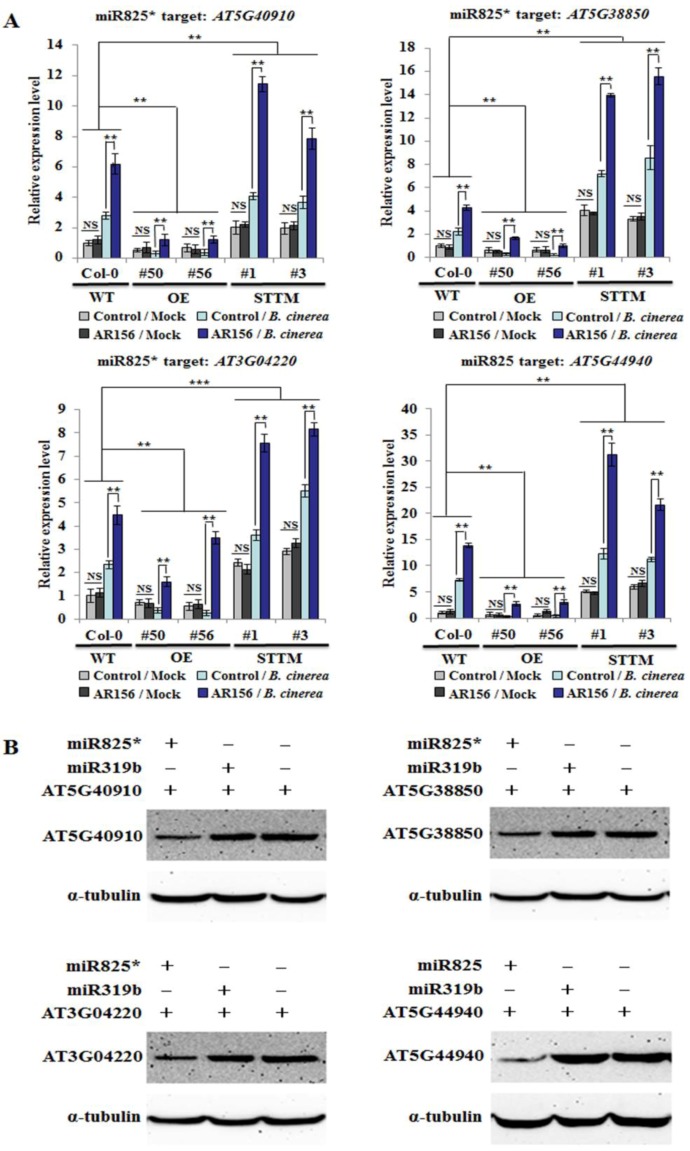
Theexpression level of target genes of miR825 and miR825* in different genotypes. (**A**) Expression profiles of *AT5G40910*, *AT5G38850, AT3G04220* (miR825* targets), and *AT5G44940* (a miR825 target) in Col-0, miR825/825* OE lines (#50 and #56), and STTM825/825* transgenic lines (#1 and #3) during AR156-triggered ISR. Three-week-old plants were pretreated with AR156 at 5 × 10^8^ CFU/mL or 0.85% NaCl (the control) as a root drench; at 7 dpt, leaves were challenge-inoculated as depicted above. At 48 hpi, total RNA was extracted from leaves. The data are presented as mean ± SD from three biological replicates. Two-way ANOVA indicated significant effects of genotype on the expression of *AT5G40910*, *AT5G38850*, and *AT5G44940* (*P* < 0.01 for the differences between WT and OE, and also for those between WT and STTM) and on *AT3G04220* expression (*P* < 0.01 and *P* < 0.001 for the differences between WT and OE, and between WT and STTM, respectively); and a significant treatment effect (*P* < 0.01) and a significant interactive effect (*P* < 0.01) on the expression of each tested gene. Note: NS, not significant; ** *P* < 0.01; *** *P* < 0.001. (**B**) Co-expression of miR825/miR825* and the four target proteins (AT5G40910, AT5G38850, AT3G04220, and AT5G44940) in a transient expression system analyzed by Western blotting. Through *Agrobacterium*-mediated transformation, the target proteins and miR825/825* or an irrelevant miRNA (miR319b) were co-expressed in *Nicotiana benthmiana* and detected at 2 dpi by using an anti-HA antibody. Here, α-tubulin served as an equal loading control. The assays were repeated three times, with similar results obtained.

**Figure 6 ijms-20-05032-f006:**
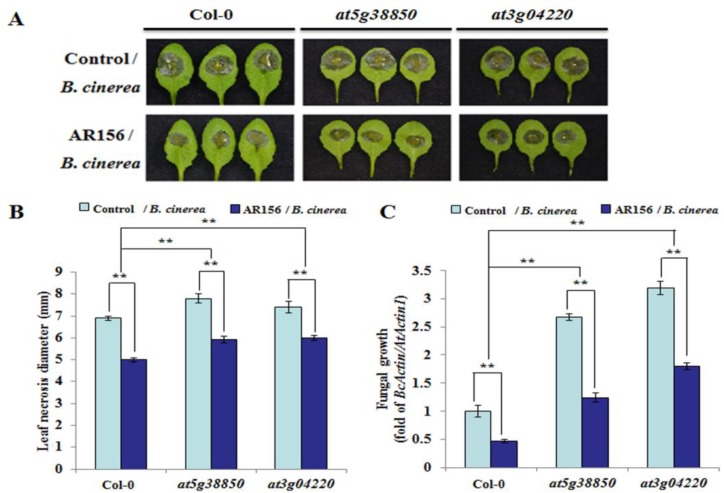
The phenotype ofplants silencing miR825/825* target genes by *B. cinerea* infection. Three-week-oldplants of *Arabidopsis* Col-0 and miR825/825* target mutant lines (*at5g38850*, *at3g04220*) were pretreated with AR156 at 5 × 10^8^ CFU/mL or 0.85% NaCl (the control) as a root drench. All plants were challenge-inoculated at 7 dpt as described above. (**A**) Disease symptoms developed at 2 dpi on leaves of each tested plant line. (**B**) Necrotic lesions were assessed at 2 dpi by measuring their average diameter on one leaf per plant, with a total of three plants assessed per sample. Two-way ANOVA showed significant effects of genotype (*P* < 0.01) and treatment (*P* < 0.01) and a significant interactive effect (*P* < 0.01). Note: ** *P* < 0.01. (**C**) In planta fungal growthin leaves of each tested plant line. The biomass of *B. cinerea* B1301 was measured as depicted above and indicated as *BcActin/AtActin1*. Two-way ANOVA indicated significant effects of genotype (*P* < 0.01) and treatment (*P* < 0.01) and a significant interactive effect (*P* < 0.01). Note: ** *P* < 0.01. The data are presented as mean ± SD from three biological replicates. The assays were repeated three times,yielding similar results.

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
