# Peer review of "Function of miR825 and miR825* as Negative Regulators in Bacillus cereus AR156-elicited Systemic Resistance to Botrytis cinerea in Arabidopsis thaliana"

_ijms, 2019, doi:10.3390/ijms20205032_

Round 1

Reviewer 1 Report

The manuscript entitled “miR825 and miR825* function as negative regulators in Bacillus cereus AR156-elicited systemic resistance to Botrytis cinerea in Arabidopsis thaliana” describe the possible role of two miRNA in the resistance of plants against Botrytis cinerea. The results presented are really interesting and the manuscript is reasonably well written. Despite this, some aspects could be addressed in order to improve the quality of the manuscript. As a general comment, the manuscript would benefit from a revisión of the English grammar and expression.

Specific comments:

Line 110: in section 2.1. Did the authors measured the level of disease observed in the figure 1A?

Line 117: Please, remove the “of” between Both and non-treated

Line 140 and the following figure captions: please consider to change the first sentence of the figure captions to a brief description of the figure instead of a description of the results.

Line 174: The authors describe the “fungal growth rate” as the ratio between BcActin/AtActin1. However, the term “fungal growth rate” is usually referred to the size of mycelium or lesion per unit of time. Please, consider changing this term by “Fungal biomass”

Line 199: The term reporter genes is usually reserved for GFP, GUS in transgenic constructions. Please, use the term marker genes or response genes instead.

Line 504 to 509: these lines are already described above, there is no need to repeat it.  

Author Response

Reviewer 1: Comments to the Author The manuscript entitled “miR825 and miR825* function as negative regulators in Bacillus cereus AR156-elicited systemic resistance to Botrytis cinerea in Arabidopsis thaliana” describe the possible role of two miRNA in the resistance of plants against Botrytis cinerea. The results presented are really interesting and the manuscript is reasonably well written. Despite this, some aspects could be addressed in order to improve the quality of the manuscript. As a general comment, the manuscript would benefit from a revision of the English grammar and expression. Specific comments: 1)Comment: Line 112: in section 2.1. Did the authors measured the level of disease observed in the Figure 1A? Response: We measured the level of Col-0 (WT) disease observed in the Figure 2A-C, including leaf phenotype (Figure 2A), leaf necrosis diameter (Figure 2B) and fungal biomass (Figure 2C). 2)Comment: Line 119: Please, remove the “of” between Both and non-treated. Response: We’re grateful to the reviewer for the advice. We have removed the “of” between Both and non-treated in Line 115. 3)Comment: Line 141 and the following figure captions: please consider to change the first sentence of the figure captions to a brief description of the figure instead of a description of the results. Response: As the reviewer’s suggestion, we have changed the first sentence of the figure captions to a brief description of the figure instead of a description of the results in the new manuscript. 4)Comment: Line 172, 173, 175, 190, 192, 348, 352, 368, 474, and 476: The authors describe the “fungal growth rate” as the ratio between BcActin/AtActin1. However, the term “fungal growth rate” is usually referred to the size of mycelium or lesion per unit of time. Please, consider changing this term by “Fungal biomass”. Response: We thank the reviewer for the suggestion. We have changed fungal growth rate to fungal biomass in Line 175, 177, 179, 195, 197, 355, 359, 376, 487, and 489. 5)Comment: Line 200: The term reporter genes is usually reserved for GFP, GUS in transgenic constructions. Please, use the term marker genes or response genes instead. Response: We thank the reviewer for pointing out this problem. We have changed “reporter genes” to “response genes” in Line 205. 6)Comment: Line 501 to 506: these lines are already described above, there is no need to repeat it. Response: We thank the reviewer for pointing out this problem. We have removed the repeat section in Line 513 to 517.

Reviewer 2 Report

Biological control is an environment friendly strategy for protecting plants from pathogen infection with little side effects. The plant-growth-promoting rhizobacteria strain AR156 can trigger ISR against Pst DC3000 in Arabidopsis and can be an essential candidate strain for biological control.

In this manuscript, the authors investigated the regulation and function of miR825/825* during AR156 elicited ISR reaction. AR156 treatment down-regulate the expression of miR825/825*. Transgenic Arabidopsis lines with suppressed miR825/miR825* expression were more resistant to B. cinerea B1301 while the Arabidopsis lines with the overexpression of miR825/miR825* were more susceptible. With two-away ANOVA test, significant interactive effects of miR825/miR825* expression level and AR156 treatment were found for the expression of defense-related genes involved in ISR. In addition, the accumulation of hydrogen peroxide, callose deposition and MAPK3/6 kinases activities confirmed the involvement of miR825/miR825* in ISR reaction. Further research revealed that the interactive effects were probably due to 4 miR825/miR825* target genes. Silencing of the miR825/miR825*-targeted genes caused higher susceptibility to B. cinera, which demonstrated significant interactive effect with the AR156. It is proposed that the AR156-triggered ISR can protect plants form B. cinerea B1301 by down-regulating miR825/miR825* to enhance the expression of the targeted defense genes. This manuscript provided fundamental knowledge and valuable data for understanding the mechanism and utilizing the essential biological control strains for the improvement of plant resistance to varied pathogens. If the authors could address the issues below, the manuscript would meet the high standard for publication.

Main issues:

It would be better to include a diagram illustrating the strategies for miR825/miR825* over-expression and suppression with STTM. In terms of the language of the text, several grammar and format mistakes need to be corrected. I listed several of them below and the manuscript needs focused, discreet proofreading.

Some other issues:

Line 24, “… more susceptible to the pathogen”. Line 104, do you mean “… in a similar manner during AR156-triggered ISR as that of defense-related genes”? Line 122-126, “To investigate whether miR825 and miR825* have function in the mediation of AR156-triggered ISR 122 against B. cinerea B1301, we examined the expression of the two miRNAs in AR156/B. cinerea, Control/B. cinerea, AR156/Mock, and Control/Mock at 48 hours post-inoculation (hpi).” Line 128-131, do you mean” B. cinerea infection rather than AR156 treatment significantly suppressed the expression of miR825*, and AR156 pretreatment accelerated the extent of the downregulation caused by B. cinerea.”? Line 132-136, “There was no difference of miR825 expression observed between AR156/Mock and Control/Mock, which was similar to the expression profile of miR825*; whereas miR825 expression was also downregulated upon B. cinerea B1301 infection in the control plants, with the downregulation significantly stronger in AR156-treated plants (Figure 1B).”. Line 136-138, “The results indicated that AR156 pretreatment resulted in significant suppression of the expression of miR825 and miR825*, especially during B. cinerea infection, suggesting that they might function as negative regulators of AR156-mediated ISR to prevent B. cinerea B1301.”. Line 156, “…in AR156-treated and non-treated plants.” Line 162-165, it would be better to move the Phytophthora and CMV part to just before the part for RT-PCR. Line 244, “…; and the intensity increased…” Line 247-248, “Notably, the phosphorylation of the two kinases sustained…”. Line 249, “…, with the intensity…” Line 250, “…throughout the whole time course (Figure 4C)”. Line 253, “To further test the hypothesis,…”. Line 282-284, “…with or without B. cinerea B1301 inoculation. The expression of the targeted genes in plants…”.

Author Response

Reviewer 2: Comments to the Author Biological control is an environment friendly strategy for protecting plants from pathogen infection with little side effects. The plant-growth-promoting rhizobacteria strain AR156 can trigger ISR against Pst DC3000 in Arabidopsis and can be an essential candidate strain for biological control. In this manuscript, the authors investigated the regulation and function of miR825/825* during AR156 elicited ISR reaction. AR156 treatment down-regulate the expression of miR825/825*. Transgenic Arabidopsis lines with suppressed miR825/miR825* expression were more resistant to B. cinerea B1301 while the Arabidopsis lines with the overexpression of miR825/miR825* were more susceptible. With two-away ANOVA test, significant interactive effects of miR825/miR825* expression level and AR156 treatment were found for the expression of defense-related genes involved in ISR. In addition, the accumulation of hydrogen peroxide, callose deposition and MAPK3/6 kinases activities confirmed the involvement of miR825/miR825* in ISR reaction. Further research revealed that the interactive effects were probably due to 4 miR825/miR825* target genes. Silencing of the miR825/miR825*-targeted genes caused higher susceptibility to B. cinerea, which demonstrated significant interactive effect with the AR156. It is proposed that the AR156-triggered ISR can protect plants form B. cinerea B1301 by down-regulating miR825/miR825* to enhance the expression of the targeted defense genes. This manuscript provided fundamental knowledge and valuable data for understanding the mechanism and utilizing the essential biological control strains for the improvement of plant resistance to varied pathogens. If the authors could address the issues below, the manuscript would meet the high standard for publication. Comment: Main issues: 1) It would be better to include a diagram illustrating the strategies for miR825/miR825* over-expression and suppression with STTM. In terms of the language of the text, several grammar and format mistakes need to be corrected. I listed several of them below and the manuscript needs focused, discreet proofreading. Response: We generated the miR825/miR825* over-expression and suppression with STTM lines in our previous study (Niu et al., 2016). To generate the overexpression construct of miR825 and miR825*, the miR825 precursors were cloned using a miR319 backbone (based on Web MicroRNA Designer) into a Gateway destination vector, pEG100 (Earley et al. 2006; Schwab et al. 2006). The STTM825 and 825*constructs used to inactivate miR825 and miR825* were generated according to Yan and colleagues (Yan et al., 2012). All constructs described were electroporated into Agrobacterium tumefaciens GV3101, and used to transform Arabidopsis by the floral dipping method (Niu et al., 2016). We also cited this paper in our Materials and Methods section (line 467). Some other issues: Comment: 1) Line 24, “… more susceptible to the pathogen”. Response: We thank the reviewer for pointing out this problem. We have changed “more prone to it” to “more susceptible to the pathogen” in Line 22. Comment: 2) Line 106, do you mean “… in a similar manner during AR156-triggered ISR as that of defense-related genes”? Response: Yes, target genes of miR825 and miR825* are expressed in a similar manner in AR156-triggered ISR. Comment: 3) Line 122-126, “To investigate whether miR825 and miR825* have function in the mediation of AR156-triggered ISR 122 against B. cinerea B1301, we examined the expression of the two miRNAs in AR156/B. cinerea, Control/B. cinerea, AR156/Mock, and Control/Mock at 48 hours post-inoculation (hpi).” Response: We have revised it as the suggestion (line 120-124). Comment: 4) Line 128-131, do you mean “B. cinerea infection rather than AR156 treatment significantly suppressed the expression of miR825*, and AR156 pretreatment accelerated the extent of the downregulation caused by B. cinerea.”? Response: Yes, AR156 pretreatment accelerated the extent of the downregulation caused by B. cinerea. Comment: 5) Line 133-136, “There was no difference of miR825 expression observed between AR156/Mock and Control/Mock, which was similar to the expression profile of miR825*; whereas miR825 expression was also downregulated upon B. cinerea B1301 infection in the control plants, with the downregulation significantly stronger in AR156-treated plants (Figure 1B).”. Response: We have revised it as the suggestion (line 130-134). Comment: 6) Line 136-138, “The results indicated that AR156 pretreatment resulted in significant suppression of the expression of miR825 and miR825*, especially during B. cinerea infection, suggesting that they might function as negative regulators of AR156-mediated ISR to prevent B. cinerea B1301.”. Response: We have revised it as the suggestion (line 136). Comment: 7) Line 157, “…in AR156-treated and non-treated plants.” Response: We have revised it as the suggestion (line 156). Comment: 8) Line 162-165, it would be better to move the Phytophthora and CMV part to just before the part for RT-PCR. Response: We’re grateful to the reviewer for the good advice. We have moved the Phytophthora and CMV part to just before the part for RT-PCR in Line 170-174. Comment: 9) Line 246, “…; and the intensity increased…” Response: We have revised it as the suggestion in Line 250. Comment: 10) Line 249-250, “Notably, the phosphorylation of the two kinases sustained…”. Response: We have revised it as the suggestion in Line 254 . Comment: 11) Line 251, “…, with the intensity…” Response: We have revised it as the suggestion in Line 255. Comment: 12) Line 252, “…throughout the whole time course (Figure 4C)”. Response: We have revised it as the suggestion in Line 256. Comment: 13) Line 255, “To further test the hypothesis,…”. Response: We have revised it as the suggestion in Line 259. Comment: 14) Line 284-286, “…with or without B. cinerea B1301 inoculation. The expression of the targeted genes in plants…”. Response: We’re grateful to the reviewer for the good advice. We have deleted “We predicted that miR825/825* targets and their cognate miRNAs would exhibit opposite expression patterns in ISR-expressing plants.” in Line 290-292.

Reviewer 3 Report

The authors showed that B. cereus AR156-induced ISR was negatively regulated by miR825 and miR825* using overexpression and knock-down (STTM) lines of miR825 and miR825* in A. thaliana. The effects of miR825/miR825* expression on ISR were assessed by disease development, fungal growth, and indicators of immune response (activation of PR genes expression, MAPK phosphorylation, expression of PTI marker genes, H2O2 generation, and callose deposition). Predicted target genes appears to be targeted by miR825 and miR825*, and mutants of their genes showed enhanced disease susceptibility. Alteration of miR825 and miR825* expressions also affected virulence of the other pathogens, oomycetes P. capsici and CMV in a similar manner. This work is interesting and appears to have been carried out soundly and the results are clear. However, I have some comments as follows.

1) Fig1B: The down-regulation of miR825 and miR825* was observed in B. cinerea-inoculated tissue, but not in systemic tissue in mock-inoculated plants primed by B. cereus AR156, indicating that miR825 and miR825* expressions were regulated in response to B. cinereal infection itself. Were their miRNAs also down-regulated in response to P. capsici and CMV infection?

2) Fig.5: miR825 and miR825* appears to target multiple transcripts of R genes and their mutants showed slightly enhanced disease susceptibility. Please discuss how R genes involve in ISR induction because, in general, R protein is specifically activated when the corresponding avr protein is recognized.

3) Fig.5B: To gain solid conclusion, the authors should show direct evidence indicating the cleavage of target transcripts by miR825 and/or miR825* in vivo in A. thaliana, not just in heterologous experimental condition in N. benthamiana.

4) Previous study showed that miR825 and miR825* negatively regulate ISR induced by B. cereus AR156 against Pseudomonas syringae (Niu et al., Journal of Integrative Plant Biology, 2016), and, in this study, authors showed miR825 and miR825* also regulated ISR against B. cinerea, P. capsici, and CMV. Please discuss how miR825 and miR825*-involving pathway expresses broad spectrum defense. Is it mediated by auxin pathway or by activation of multiple R proteins?

Author Response

Reviewer 3: Comments to the Author The authors showed that B. cereus AR156-induced ISR was negatively regulated by miR825 and miR825* using overexpression and knock-down (STTM) lines of miR825 and miR825* in A. thaliana. The effects of miR825/miR825* expression on ISR were assessed by disease development, fungal growth, and indicators of immune response (activation of PR genes expression, MAPK phosphorylation, expression of PTI marker genes, H2O2 generation, and callose deposition). Predicted target genes appears to be targeted by miR825 and miR825*, and mutants of their genes showed enhanced disease susceptibility. Alteration of miR825 and miR825* expressions also affected virulence of the other pathogens, oomycetes P. capsici and CMV in a similar manner. This work is interesting and appears to have been carried out soundly and the results are clear. However, I have some comments as follows. Comment: 1) Fig1B: The down-regulation of miR825 and miR825* was observed in B. cinerea-inoculated tissue, but not in systemic tissue in mock-inoculated plants primed by B. cereus AR156, indicating that miR825 and miR825* expressions were regulated in response to B. cinerea infection itself. Were their miRNAs also down-regulated in response to P. capsici and CMV infection? Response: The down-regulation of miR825 and miR825* was observed in B. cinerea-inoculated tissue, but not in systemic tissue in mock-inoculated plants primed by B. cereus AR156, B. cinerea infection with AR156-pretreatment significantly suppressed the expression of miR825*, and AR156 pretreatment accelerated the extent of the downregulation caused by B. cinerea infection, suggesting that they might function as negative regulators of AR156-mediated ISR to prevent B. cinerea B1301. We do not detect the miR825 and miR825* expression level in response to P. capsici and CMV. Comment: 2) Fig.5: miR825 and miR825* appears to target multiple transcripts of R genes and their mutants showed slightly enhanced disease susceptibility. Please discuss how R genes involve in ISR induction because, in general, R protein is specifically activated when the corresponding avr protein is recognized. Response: miR825 and miR825* expression was downregulated upon B. cinerea B1301 infection in the control plants, with the downregulation significantly stronger in AR156 pretreatment and B1301 infection plants, suggesting that they might function as negative regulators of AR156-mediated ISR to prevent B. cinerea B1301. Moreover, miR825 and miR825* target multiple transcripts of R genes, our study demonstrated that miR825 and miR825* act as negative regulators of AR156-triggered ISR to prevent B. cinerea B1301 by repressing the transcription of the targeted defense genes activated by AR156 in plants upon its attack. Therefore, we infer that these R genes may involve in AR156-induced ISR. Comment: 3) Fig.5B: To gain solid conclusion, the authors should show direct evidence indicating the cleavage of target transcripts by miR825 and/or miR825* in vivo in A. thaliana, not just in heterologous experimental condition in N. benthamiana. Response: We’re grateful to the reviewer for the suggestion. To investigate whether miR825 and miR825* modulate ISR by affecting the expression of their targets genes, qRT-PCR was conducted to examine expression patterns of miR825/825* target genes in Col-0, miR825/825* OE lines, and STTM825/825* lines. To further confirm that these genes are targeted by miR825 and miR825*, the two miRNAs were separately transiently co-expressed with a target gene fused with the HA-tag in Nicotiana benthamiana. We will examine the cleavage of target transcripts by miR825 and/or miR825* in vivo in A. thaliana in the future study. Comment: 4) Previous study showed that miR825 and miR825* negatively regulate ISR induced by B. cereus AR156 against Pseudomonas syringae (Niu et al., Journal of Integrative Plant Biology, 2016), and, in this study, authors showed miR825 and miR825* also regulated ISR against B. cinerea, P. capsici, and CMV. Please discuss how miR825 and miR825*-involving pathway expresses spectrum defense. Is it mediated by auxin pathway or by activation of multiple R proteins? Response: we added more discussion in the manuscript (line 441-449). Many studies showed that R genes-involving pathway expresse spectrum defense (Glowacki et al., 2010). In some cases, correlation between resistance and induction of R genes, or reduction of miRNA and miRNA-triggered phasiRNA, was observed among different crop varieties (Kundu et al., 2017; Ouyang et al., 2014; Yang et al., 2015). In Arabidopsis, the miR472-RDR6 pathway was shown to negatively regulate both PTI and ETI constitutively and inhibit induction of R genes during infection (Boccara et al., 2014). Besides biotic stress, various abiotic stresses are also shown to modulate miRNA-mediated R gene regulation. It is reported that in Populus trichocarpa, miR482 accumulation in developing xylem was decreased by tension and compression stress (Lu et al., 2005). miRNA-mediated regulation of plant R genes emerges as an impotant general mechanism operating in all R gene-encoding plants examined to date (Deng et al., 2018). In our study, miR825 and miR825* target R genes, therefore, we think that miR825 and miR825*-involving in AR156-induced ISR expresses spectrum defense, which is mediated by activating multiple R proteins. The further study is needed in the further.

Round 2

Reviewer 3 Report

I have no more concern with the manuscript.